# Arthropods as Vectors of Grapevine Trunk Disease Pathogens: Quantification of *Phaeomoniella chlamydospora* on Arthropods and Mycobiome Analysis of Earwig Exoskeletons

**DOI:** 10.3390/jof10040237

**Published:** 2024-03-22

**Authors:** Elisa Maria Brandenburg, Ralf Thomas Voegele, Michael Fischer, Falk Hubertus Behrens

**Affiliations:** 1Julius Kühn-Institute (JKI), Institute for Plant Protection in Fruit Crops and Viticulture, 76833 Siebeldingen, Germany; michael.fischer@julius-kuehn.de (M.F.); falk.behrens@julius-kuehn.de (F.H.B.); 2Department of Phytopathology, Institute of Phytomedicine, Faculty of Agricultural Sciences, University of Hohenheim, 70599 Stuttgart, Germany; ralf.voegele@uni-hohenheim.de

**Keywords:** grapevine trunk diseases, esca, arthropods, vectors, dispersal, spore loads, earwigs, mycobiome

## Abstract

Viticulture worldwide is challenged by grapevine trunk diseases (GTDs). Involvement of arthropods in the dissemination process of GTD pathogens, notably esca pathogens, is indicated after detection of associated pathogens on arthropod exoskeletons, and demonstration of transmission under artificial conditions. The present study is the first to quantify spore loads via qPCR of the esca-relevant pathogen *Phaeomoniella chlamydospora* on arthropods collected in German vineyards, i.e., European earwigs (*Forficula auricularia*), ants (*Formicidae*), and two species of jumping spiders (*Marpissa muscosa* and *Synageles venator*). Quantification of spore loads showed acquisition on exoskeletons, but most arthropods carried only low amounts. The mycobiome on earwig exoskeletons was described for the first time to reveal involvement of earwigs in the dispersal of GTDs in general. Metabarcoding data support the potential risk of earwigs as vectors for predominantly *Pa. chlamydospora* and possibly *Eutypa lata* (causative agent of Eutypa dieback), as respective operational taxonomical unit (OTU) assigned genera had relative abundances of 6.6% and 2.8% in total reads, even though with great variation between samples. Seven further GTD-related genera were present at a very low level. As various factors influence the successful transmission of GTD pathogens, we hypothesize that arthropods might irregularly act as direct vectors. Our results highlight the importance of minimizing and protecting pruning wounds in the field.

## 1. Introduction

Various fungal diseases constitute an important threat for viticulture worldwide. The impact of grapevine trunk diseases (GTDs) significantly increased in recent decades [1]. The esca disease complex as well as Botryosphaeria and Eutypa dieback are the three main GTDs in established vineyards [2,3,4,5]. Causal agents are phytopathogenic fungi that internally colonize grapevine wood, deteriorating the vascular tissue, and eventually decreasing the vitality and productivity of infested vines [3,6,7,8].

Eutypa dieback, also known as eutypiosis, appears as cankers on trunks and vine arms, discolorations and v-shaped necrosis in the vascular tissue, stunted shoots, necrotic or short leaves, wilted flowers, and malformed berries. Causal agents belong to the family Diatrypaceae, with *Eutypa lata* being prevalent and most crucial [9,10,11].

Vines affected by Botryosphaeria dieback, also referred to as “black dead arm” (BDA), show discolorations of the vascular tissue, premature leaf fall, dieback of shoots, and withering of fruits and flowers [12,13]. Responsible pathogens are members of the fungal family Botryosphaeriaceae, such as the genera *Botryosphaeria*, *Diplodia*, or *Dothiorella* [14].

Characteristic inner symptoms of the esca disease complex are discolorations of the vine’s vascular system, such as formation of gummosis and necrosis, known as “brown wood streaking” or Petri disease. External symptoms are stunted growth, the characteristic “tiger-stripe” leaf patterns called “Grapevine Leaf Stripe Disease” (GLSD), shrinking of berries, and the sudden withering of parts of or the whole vine (apoplexy) [3,4,15,16,17]. White rot frequently co-occurs with GLSD in old vines [17] and is called “esca proper”. Esca’s main causal agents are the vascular pathogenic ascomycetes *Phaeomoniella chlamydospora* [18], members of the genera *Phaeoacremonium* [19,20,21], and *Cadophora* [22,23,24], as well as wood-decaying basidiomycetes (white rot fungi) belonging to the genus *Fomitiporia*, mainly *F. mediterranea* [25,26,27].

Symptoms of different GTDs can overlap due to the possibility of co-occurrence of involved fungal pathogens [1].

Propagules of GTD pathogens are considered as being mainly dispersed by wind or rain splash [28,29], and enter the vines through wounds in the grapevine wood, caused by winter pruning or management practices during the vegetative season [30,31,32,33,34]. Susceptibility of pruning wounds to pathogen infection can remain for several weeks, and is influenced by various factors, such as the fungal pathogen invading, grapevine variety, and pruning season [34].

As is known from other plant diseases, arthropods can play a role as vectors of phytopathogenic fungi by transporting fungal spores externally or internally to potential infection sites [35,36,37,38,39,40]. Involvement of arthropods in the dissemination process of GTDs, primarily esca, has been previously described. Besides the spatial distribution of propagules on and between grapevines, direct transmission of pathogen inoculum to susceptible wounds by arthropod vectors is possible [41,42,43]. The potential of arthropod-mediated transmission of esca-associated pathogens was, however, only evaluated under artificial conditions [43,44]. In order to evaluate the importance of spore loads in the field, the actual quantity of GTD-associated fungal material on arthropod exoskeletons ought to be determined.

The European earwig *Forficula auricularia* (Dermaptera: Forficulidae), one predominant arthropod species in German vineyards [42,45], has been shown to potentially vector esca pathogens, provided that the susceptibility of grapevine wounds is given [42,43]. Earwigs collected from vineyards frequently carried esca-associated pathogens on their exoskeletons, and the simultaneous detection of different esca pathogens was also evident [42]. Furthermore, earwig-mediated transmission of esca-associated pathogens was successful under artificial conditions leading to the infection of healthy vines [43]. In addition, the detection of fungi associated with Botryosphaeria and Eutypa dieback from earwig exoskeletons underline the possible involvement of earwigs in the dissemination process of GTDs [41]. In the present study, next-generation sequencing (NGS) is used as a culture-independent method to elucidate earwigs’ involvement at dispersing fungal diseases of grapevines by displaying the mycobiome present on their exoskeletons.

The aims of the present study were as follows: (i) quantifying *Pa. chlamydospora* on the selected arthropod species collected in German vineyards using qPCR, and (ii) analyzing the mycobiome of earwig exoskeletons using NGS to determine the spectrum of GTD-related fungi.

Our results aim to gain insights into the possible contribution of arthropods in the dissemination process of GTD-related pathogens in the field. Based on our results, we highlight the importance of reducing and protecting grapevine pruning wounds in order to minimize pathogen invasion.

## 2. Materials and Methods

### 2.1. Sampling Site and Collection of Arthropods

Arthropods were collected in 2019 and 2020 from experimental vineyards located at the Julius Kühn-Institute (JKI) in Siebeldingen, Germany. Vineyard “A” (49°13′00.2″ N; 8°02′53.1″ E) was planted in 1996 with the fungus-resistant (PIWI-) cultivar *Vitis vinifera* cv. ‘Phoenix’ (resistance against powdery and downy mildew) and was managed with integrated plant protection. Vineyard “B” (49°13′08.8″ N; 8°02′39.6″ E), planted in 2002, comprised two PIWI- (‘Calandro’ and ‘Regent’) and two traditional cultivars (‘Pinot Noir’ and ‘Riesling’), and the plant protection measures were organic. In the years 2012 to 2015, GLSD-symptoms and apoplexy were frequently observed in vineyard “A” [46], whereas symptoms were rare in vineyard “B”.

Cardboard-traps were used to collect arthropods directly from selected vines (for details see [42]). According to the period of arthropod activity observed in the vineyards, traps were installed in early April and arthropods were collected until the end of October. Collection was carried out by emptying traps every other week into a large plastic box and aseptically placing individual arthropods into 2 mL reaction tubes. The mass occurrence of earwigs was handled by considering five randomly chosen individuals as one sample, with a maximum of five such samples per trap. The morphological identification of arthropod species was accomplished using an identification guide for German fauna [47] and a field guide for spiders [48].

### 2.2. Washing of Arthropod Exoskeletons and DNA Extraction

Esca-related pathogens on arthropod exoskeletons were detected by following the protocol described in Kalvelage et al. [42]. Arthropods were freeze-killed prior to washing off potential spores from their exoskeletons. Genomic DNA (gDNA) was extracted from these washing suspensions.

### 2.3. qPCR for the Quantification of Pa. chlamydospora on Arthropods

#### 2.3.1. Selection of Arthropod Samples for qPCR Analysis

A nested multiplex PCR of gDNA samples obtained from arthropod washing suspensions showed the presence of the esca-related pathogens *Pa. chlamydospora*, *C. luteo-olivacea,* and *Phaeoacremonium* spp. [42]. From this set, a total of 329 samples obtained from earwigs, two jumping spiders (*Marpissa muscosa* and *Synageles venator*), and ants that contained *Pa. chlamydospora* spores were selected for qPCR analysis as described below.

#### 2.3.2. Construction of the Standard Curve

A representative isolate of *Pa. chlamydospora* also used in the transmission experiments conducted by Kalvelage et al. [43] was grown on potato dextrose agar (PDA; Carl Roth GmbH + Co., KG, Karlsruhe, Germany) plates for four weeks at room temperature under daylight conditions. A conidial suspension was prepared by detaching conidia from the medium using 3 mL of sterile water and filtering the suspension through an ADVANTEC^®^ membrane filter with 5 µm pore size (Toyo Roshi Kaisha, Ltd., Tokyo, Japan). The concentration was determined using a hemocytometer (Neubauer improved). A 10-fold dilution series was made from the initial spore suspension; gDNA was extracted [42] and used as standard for the qPCR as described below. The standard regression curve was generated by plotting the logarithm with base 10 of the known spore concentrations (conidia/µL) against the threshold cycle (C_T_) values measured using the 7500/7500 Fast Real-Time PCR System (Applied Biosystems, Darmstadt, Germany) (Figure A1).

#### 2.3.3. qPCR Set Up

qPCR was carried out with Luna^®^ Universal (RT)-qPCR reagents (New England Biolabs, Ipswich, MA, USA) and a 7500/7500 Fast Real-Time PCR System (Applied Biosystems). The *Pa. chlamydospora* specific primers Pa. chlamydospora1H: 5′-CCC GAT CTC CAA CCC TTT GTT T-3′ and Pa. chlamydospora2H: 5′-CGG GCC TAT CTT CTA TGA AAG-3′ [49] were used.

The 20 µL PCR reactions were set up as follows: 1 µL template gDNA; 0.25 µM of each primer; 10 µL Luna Universal qPCR Mix; adjusted with “Bioscience-grade”-water (Carl Roth GmbH + Co., KG). Cycle conditions of the 2-step protocol were as follows: warming-up phase at 50 °C for 2 min, holding stage at 95 °C for 10 min, and 40 cycles for 15 s at 95 °C and 30 s at 60 °C. Melting curves were examined to verify the specific amplifications.

#### 2.3.4. qPCR Sample Processing

In each run, 28 different gDNA samples and a standard dilution series were analyzed in three replicates. The final standard curve comprised 2.23 × 10^5^ to 2.23 × 10^2^ spores, with this spanning the number of spores observed in the examined samples. A no-template control was run in two replicates. The linear regression equation of the standards was used to estimate the conidial concentration in the arthropod washing suspensions. The number of individuals was considered for pooled earwig samples.

As 223 spores were determined as the detection limit for conidia counts, the respective C_T_ value in each qPCR round was set as threshold for the cutoff of false-positive results. The number of *Pa. chlamydospora* spores on arthropod exoskeletons was analyzed using a linear mixed effect model (lmer) with log2-transformed values. Fixed factors were species and year and their interaction, and the intercept per season:vineyard was considered as a random effect. Based on this model, an ANOVA (Type II Wald chi-square test) was calculated.

### 2.4. Mycobiome Analysis of Earwig Exoskeletons Using NGS

#### 2.4.1. Selection of Earwig Samples for DNA Metabarcoding

A total of 120 randomly selected gDNA samples from earwig exoskeletons obtained from June to August each in 2019 and 2020, originally proven to be positive or negative for *Pa. chlamydospora* [42], were subjected to an ITS end point PCR after quality control regarding the DNA concentration needed for metabarcoding.

The primers used are ITS1catta: 5′-ACCWGCGGARGGATCATTA-3′ and ITS2ngs: 5′-TTYRCKRCGTTCTTCATCG-3′ being specific for the fungal ITS1 region [50]. Both primers had a final concentration of 0.3 µM. The PCR reaction was carried out in a SimpliAmp™ Thermal Cycler (Applied Biosystems) and the reaction was conducted with the KAPA HiFi Hotstart Taq polymerase (Peqlab, Erlangen, Germany). The reaction volume was 20 µL, according to the user manual, and 1 µL of the extracted DNA was used as template. PCR conditions were as follows: initial denaturation at 95 °C for 3 min; 30 cycles of 30 s at 98 °C, 20 s at 54 °C, and 20 s at 72 °C; final extension for 1 min at 72 °C. PCR products were loaded on a 1.5% agarose gel, run at 6 V/5 cm for 45 min, and visualized under ultraviolet (UV) light using a QUANTUM ST5 gel documentation system (Vilber Lourmat, Eberhardzell, Germany).

#### 2.4.2. DNA Metabarcoding

AllGenetics & Biology SL (A Coruña, Spain) created sequence libraries and performed high-throughput sequencing for metabarcoding. The primers ITS1catta and ITS2ngs amplifying the fungal ITS1 region [50] were used to create DNA libraries. The 25 µL primary PCR reaction was composed as follows: 2.5 µL DNA; 0.5 µM of each primer; 12.5 µL Supreme NZYTaq 2× Green Master Mix (NZYTech, Lisboa, Portugal); adjusted with “Bioscience-grade”-water. The PCR conditions were as follows: initial denaturation at 95 °C for 5 min, 35 cycles at 95 °C for 30 s, 48 °C for 45 s, 72 °C for 45 s, final elongation at 72 °C for 7 min. Oligonucleotide-index sequences were attached in a secondary PCR reaction in order to multiplex different libraries on a sequence pool. PCR conditions were set according to the first PCR reaction, but included 5 instead of 35 cycles, and the annealing temperature was increased to 60 °C. The size of PCR products was visualized on a 2% agarose-gel, stained with GreenSafe (NZYTech). Purification of PCR products was conducted using the Mag-Bind RXNPure Plus magnetic beads (Omega Biotek, Norcross, GA, USA) according to the user manual. After pooling PCR products, they were sequenced using Illumina MiSeq PE300 technology (Illumina Germany, Berlin, Germany).

Template-free library preparations were also analyzed in order to exclude contaminations. For overall quality control, an artificial mock community comprising DNA of twelve in vitro cultivated fungal species (*Aspergillus ochraceus*, *Aureobasidium pullulans*, *Bjerkandera adusta*, *Botrytis cinerea*, *Fomitiporia mediterranea*, *Fusarium avenaceum*, *Fusarium culmorum*, *Gibellulopsis nigrescens*, *Phaeoacremonium angustius*, *Phaeomoniella chlamydospora*, *Pichia kluyveri*, *Stereum hirsutum*) was additionally analyzed.

#### 2.4.3. Bioinformatic Processing

The analysis was performed using a metabarcoding pipeline only considering forward read sequences [51]. Cutadapt v3.4 [52] and the R package ShortRead v1.58.0 [53] were used to cut primers, and dada2 v1.28.0 [54] was used for quality filtering, denoising with independent sample inference and eliminating chimeras. Vsearch v2.22.1 [55] was used for post-clustering 98.5% operational taxonomical units (OTUs) from the denoised sequences, and taxonomic assignment of OTUs against the UNITE v8.2 database [56] was again performed with dada2.

Only samples with a minimum of 5000 reads were considered, and analyses were limited to OTUs belonging to the kingdoms Fungi and Stramenopila and to OTUs found in at least two samples. Relative quantification was used to describe the abundance of individual OTUs. OTUs assigned to the same genus were pooled to present the relative abundance of certain OUT-assigned taxonomic ranks in total reads.

The number of observed OTUs was analyzed using a negative binomial generalized linear mixed-effect model (glmer.nb). The sampling year was set as a fixed factor, and so was the random intercept per month; vineyard was considered as the random effect. Based on this model, an ANOVA (Type II Wald chi-square test) was calculated.

## 3. Data Analysis

Statistical analyses were conducted with the RStudio software v4.3.2 [57]. General analysis and plotting were performed with the R packages tidyverse v1.3.1 [58], ggpubr v0.4.0 [59], psych v2.2.5 [60]), lme4 v 1.1-21 [61], emmeans v1.7.4-1 [62], multcomp v1.4.19 [63], ggpubr v0.4.0 [59], and car v3.0.13 [64]. For metabarcoding data, the R packages phyloseq v1.30.0 [65], vegan v2.5.7 [66], and cowplot v1.1.1 [67] were additionally used.

## 4. Results

### 4.1. Spore Loads of Pa. chlamydospora on Arthropods

Using qPCR, spore loads of *Pa. chlamydospora* on earwigs (*F. auricularia*), ants (Formicidae), and two jumping spiders (*M. muscosa* and *S. venator*) were determined (Figure 1). A total of 59.6% of samples testing positive for *Pa. chlamydospora* in the nested PCR [42] yielded a positive result in the qPCR; vice versa, 40.4% of the samples did not contain quantifiable spore numbers. Samples that tested positive revealed great variation within each arthropod species. The effects of the species (Wald χ^2^ = 6.7726, *p* = 0.07951) and sampling year (Wald χ^2^ = 0.7906, *p* = 0.37391) were not found to affect the number of spores, but their interaction (Wald χ^2^ = 11.0965, *p* = 0.01122).

Overall, 54.3% of earwig samples contained quantifiable numbers of spores. Most earwigs carried a spore number below 10,000 on their exoskeletons. In 2020, ten outliers were found ranging from 10,000 to 40,000 spores. Of all ant samples tested in our qPCR, 41.9% contained a concentration of *Pa. chlamydospora* spores below the detection limit and the other samples revealed numbers below 5000, except for one ant carrying approx. 9000 spores. In 2019, 48.6% of *M. muscosa* individuals carried between 10,000 and 20,000 *Pa. chlamydospora* spores on their exoskeletons, whereas most spore numbers were below 5000 in 2020, and only four samples exceeded 10,000 spores. In 2019, a total of 78.6% of *S. venator* samples tested positive for *Pa. chlamydospora* and showed spore numbers between approx. 5000 and 12,500. A total of 66.6% of *S. venator* carried *Pa. chlamydospora* in 2020 with spore numbers below 2500.

### 4.2. Mycobiome Analysis of Earwig Exoskeletons

#### 4.2.1. Summary of Dataset

A total of 6,466,441 raw reads were produced from 120 samples, and 5,971,932 were retained after sequence quality control. After filtering, a total of 110 samples and 341 OTUs were retained. Sample read sums ranged from 5772 to 115,638 with a median of 55,586.5.

#### 4.2.2. Alpha Diversity Metrics

OTU richness was compared between earwig exoskeletons depending on sampling years (Figure 2). Rarefaction curves show a saturation in OTU detection for all samples, and no effect of the sampling year was found regarding the number of observed OTUs (Wald χ^2^ = 0.0045, *p* = 0.9465).

#### 4.2.3. Fungal Abundance and Diversity

The 341 retained OTUs spanned five phyla: Ascomycota, Mucoromycota, Basidiomycota (kingdom Fungi), Oomycota, Ochrophyta (kingdom Stramenopila), while some OTUs remained unidentified (Figure 3A). OTUs can further be classified into 24 classes, 63 orders, 135 families, and 191 genera.

The fifteen most frequent genera (if genus-level assignment was possible) with the highest relative abundance in total reads perceived from earwig exoskeletons are presented in Figure 3B. Determining the fungal community present on earwig exoskeletons revealed OTUs that matched with many genera associated with GTDs (for GTD-associated genera see reviews [5,68,69], presented in Figure 3C). Overall, 9.6% of the total reads were associated with fungal genera related to GTDs. OTUs with lower relative abundance are pooled in ‘Others’.

GTD-related genera *Phaeomoniella* and *Eutypa* are included in the 15 most frequent genera determined on earwig exoskeletons, and their relative abundance (total reads for every earwig sample) is presented in Figure 4. The concentration of GTD-associated pathogens differed greatly between samples and years. In 2019, 38% of samples contained *Phaeomoniella*, with relative abundances ranging up to 100%. In 2020, only 13.2% of samples contained *Phaeomoniella*, with relative abundances all below 12.5%. In only 3 out of 42 samples (7.1%), *Eutypa* was found in 2019 with relative abundances close to zero, whereas 18 out of 68 (26.5%) samples contained *Eutypa* in 2020 with relative abundances ranging up to 60%.

## 5. Discussion

The potential risk presented by arthropods as vectors of GTD pathogens, in particular esca pathogens, was highlighted with the detection of GTD-associated fungi on various arthropods collected in vineyards in South Africa and Germany [41,42]. In addition, it has been shown under artificial conditions that arthropods can transfer esca pathogens to healthy vines, causing new infections [43,44].

In the present study, qPCR was used to quantify the spore load of *Pa. chlamydospora* on certain arthropod species collected in German vineyards. As successful transmission by ants and earwigs has already been described [43,44], we focused on these taxa and additionally included two jumping spiders, as they frequently carried esca-associated pathogens in German vineyards [42]. In our qPCR analysis, we processed only samples that tested positive for *Pa. chlamydospora* in the nested PCR [42]. In total, 133 out of 329 samples (40.4%) that had been positive for *Pa. chlamydospora* in the nested PCR did not test positive in our qPCR, probably caused by a comparably lower sensitivity of the latter setup and/or target concentration below the detection limit. High variation was additionally observed between samples of the same arthropod species (Figure 1). Edwards et al. [70] compared molecular detection methods for *Pa. chlamydospora* and noted not only a lack of detection in qPCR despite positive results in nested PCR analysis, but also increasing variation with decreasing spore loads. It should also be considered that the suspension used to wash the spores off the arthropod exoskeletons [42] may not have been suitable to detach all the spores, which could also have led to fluctuations in spore numbers.

The arthropod species studied differed in size, morphology, and behavior. The following aspects could be advantageous for inoculum acquisition: a hairy surface, as is evident in jumping spiders, i.e., *M. muscosa*, to which the spores can easily adhere, and a small body size to reach inoculum deep inside or between small cracks in the vine wood [21,71]. Regarding the latter aspect, small mites have already been observed as being associated with sporulating mycelium of *Pa. chlamydospora* [71]. The main risk of arthropod-mediated transmission of spores most likely occurs during spring, when late-winter pruned vines may still be susceptible [34,72,73], and putative vectors are active and attracted to wound sap [41,42,43]. During the vegetative season, sucker and green shoot wounds may also serve as entry portals [33]. Our analysis showed that many *Pa. chlamydospora* spores are present on several arthropod species that are active in vineyards in early and late spring [42]. The average spore numbers of *Pa. chlamydospora* determined during artificial transmission experiments on ants and earwigs were 122,500 and 2,229,688, respectively [43], while another author [44] determined an average of 55,000 *Pa. chlamydospora* spores on ants. In comparison, our results revealed that 56.1% of arthropods that carried *Pa. chlamydospora* had spore numbers below 1000. The rest of the arthropod samples showed greatly varying spore numbers ranging up to 40,000, indicating that individuals irregularly came in contact with sporulating inoculum and acquired a spore package that was potentially risky for disease transmission in relation to artificial transmission experiments. Moreover, the sampling year was found to affect the number of spores indicating that the risk of arthropods as vectors may differ between years.

With regard to the results of Elena et al. [74], who considered an inoculation dose of 100–2000 spores of *Pa. chlamydospora* as sufficient for successful infection, arthropod-mediated transmission in the field might theoretically be possible. However, the molecular detection of *Pa. chlamydospora* or any other GTD pathogen on arthropods only proves its physical presence and does not allow definite statements on its viability. Because spore germination in fungi is affected by abiotic conditions such as temperature, moisture, humidity, or light [75], the actual transmission efficiency still remains unclear. In addition, the actual number of spores that eventually reach susceptible vine wounds is very likely to be much smaller than the total number of spores on arthropod exoskeletons. In our study, we used a qPCR with defined spore numbers as the standard. *Pa. chlamydospora* is a mitosporic species [18], readily producing enormous numbers of conidial spores, and these are considered the main transmission agents in the field, vectored mainly by wind and/or rain splash [28,29]. We do not know to which degree field-samples might contain mycelial fragments, which would also be testing positive in the PCR-analysis.

Samples analyzed in this study were the same as those examined by Kalvelage et al. [42] using a nested PCR approach. In total, 41.9% of the nested-positive samples did not confirm the presence of *Pa. chlamydospora* in the qPCR, which could be due not only to concentrations below the detection limit, but also to the fact that the nested PCR may have yielded some false-positive results.

The European earwig *F. auricularia* is a vector candidate for the transmission of esca pathogens in German vineyards [42,43]. In the present study, metabarcoding was used to elucidate the involvement of earwigs in the dispersal of further GTD-related fungi by describing for the first time the fungal community present on earwig exoskeletons.

The mycobiome on earwigs comprised 341 OTUs and OTU-richness can be compared to grapevine leaves in the studies conducted by Perazzolli et al. [76] and Behrens & Fischer [77], whereas 897 OTUs were detected by Castañeda et al. [78]. In comparison to the mycobiome of grapevine wood, the fungal diversity on earwigs was slightly lower compared to studies that found approx. 500 OTUs [79,80] and was much lower in comparison to Vanga et al. [81], who detected 1250 OTUs in grapevine trunks. It has to be considered that the number of OTUs highly depends on sampling and analysis methods.

Fungal genera such as *Mucor*, *Cladosporium, Alternaria*, *Aureobasidium*, *Mycosphaerella*, and other members of the family *Pleosporaceae* have not only been detected on grapevine leaves [76,77,78,82] but also in grapevine wood [83,84,85]. Other fungi present in vine trunks are also described in our study, i.e., *Arthrinium*, *Petriella*, and members of the family *Nectriaceae* [84,85]. The yeast genus *Kazachstania* has not only been found in grape must [86,87], but also on earwig exoskeletons. The genus *Acrostalagmus*, a valuable bioactive component producer [88], identified in pooled grapevine samples [89], and the genus *Neodidymelliopsis* frequently detected on xylem-feeding leafhoppers [90] were also detected in the present study.

Our analysis of earwig exoskeletons revealed the presence of fungi associated with the esca disease, such as *Phaeomoniella* [15,18], *Phaeoacremonium* [19,20,21], and *Cadophora* [22,23,24]. We also identified the genus *Eutypa* associated with Eutypa dieback [91], the genus *Diplodia* associated with Botryosphaeria dieback [13], the genus *Diaporthe* associated with Phomopsis dieback [92], and the genus *Ilyonectria* associated with black-foot disease [93].

Important esca-associated pathogens have previously been detected on earwigs using molecular techniques [41,42]. Moyo et al. [41] successfully isolated many above-mentioned GTD fungi from other arthropods collected in vineyards. In our experiments, the diversity of GTD-associated genera comprised seven different genera with a total relative abundance of 9.6%. On grapevine leaves, only *Cadophora*, *Diplodia,* and *Diaporthe* were detected [82,94]. The relative abundance of *Diplodia* ranged between 0.67% and 18% [82] and relative abundances of *Cadophora* and *Diaporthe* were below 2% in the study conducted by Pinto et al. [94]. As expected, high frequencies of GTD-associated fungi ranging from approx. 42–76% have been described in vine trunk mycobiome studies [79,81,95]. The numbers of GTD-associated genera assigned by OTUs were similar to those found in our study.

As the majority of OTUs assigned to GTD-associated genera had only minor relative abundances in total reads, we assume that the risk of earwig-mediated transmission might be negligible for most pathogens. However, the genera *Phaeomoniella* and *Eutypa* are comprised in the 15 most abundant genera, with relative OTU-abundances of 6.6% and 2.8% across all samples, respectively (Figure 3). In mycobiome studies of vine trunks, *Pa. chlamydospora* and *E. lata* have been demonstrated as the most frequent OTUs [79,81,95]. Considering the sample specific relative abundance of the respective genera (Figure 4), it is obvious that only a few earwig samples contained larger relative amounts of *Phaeomoniella* or *Eutypa,* while the majority of earwigs did not come in contact with those fungi. This is in accordance with the observations of the qPCR analysis. Interestingly, *Pa. chlamydospora* had relative abundances of approx. 100% in some samples, indicating that individual earwigs have been in intensive contact with sporulating mycelium.

Basidiomycota-assigned OTUs accounted for approx. 7% of the total reads. Interestingly, no OTU was assigned to the esca-relevant genus *Fomitiporia*, although it has frequently been detected in German vineyards both using isolation and metabarcoding techniques [79,96,97,98]. Apparently, earwigs are less involved in the dispersal of white-rot agents.

Earwigs frequently carried the genus *Aureobasidium* on their exoskeletons (4.9% of total reads). The yeast-like fungus *A. pullulans* is a prominent biological antagonist of GTDs such as *Pa. chlamydospora*, *E. lata*, and *Diplodia seriata* [4,99]. Recent studies, however, point to a more complex interaction of *A. pullulans* and GTD pathogens, such as *Pa. chlamydospora*, as the co-occurrence of these fungi *in planta* has been observed to enhance disease expression, i.e., foliar symptoms of esca [100].

Our results shed more light on the role of arthropods in the dissemination of GTDs, especially esca. Considering the number of arthropods with probably relevant numbers of *Pa. chlamydospora* spores on their exoskeletons, the proportion of GTD pathogens in the mycobiome of earwig exoskeletons, and the sporadic visit of pruning wounds by potential vectors, we suggest that the risk of arthropods as vectors of GTD pathogens in German vineyards theoretically exists but appears to be negligible in the field. In addition, the co-occurrence of winter pruning wound susceptibility and the presence of potential arthropod vectors in the field is complex [34,42], whereas the risk for sucker and green shoot wounds seems to be more relevant and might need further analysis [33]. Overall, arthropods disperse pathogenic inoculum on and between vines and may irregularly act as vectors of GTD pathogens as they visit pruning wounds in search for food such as wound sap [41,42,43]. With this background, any kind of injury in the grapevine wood should be minimized, underlining the potential of minimal pruning techniques [79,101,102]. In addition, the application of appropriate pruning wound protection is an effective measure to prevent pathogen invasion [1,3,4,102]. The use of products containing biological control agents (BCAs) such as the fungal antagonist *Trichoderma atroviride* SC1 in combination with other strategies may offer a sustainable way to establish a long-term barrier against GTD fungi [103].

## 6. Conclusions

The present study is the first in which the occurrence of *Pa. chlamydospora* spores on different arthropod species in the field has been quantified. Our results show the acquisition of spores to arthropods in the field, eventually leading to the possible transmission of spores to susceptible vine wounds.

For the first time, the mycobiome of earwig exoskeletons is presented. Although fungal diversity related to GTDs was higher on earwigs in comparison to, e.g., grapevine leaves, the majority of GTD-associated OTUs accounted for a negligible relative proportion only. A relevant occurrence was described for the genera *Phaeomoniella* and *Eutypa* which could indicate a possible involvement of earwigs in the transmission of esca and Eutypa dieback in the field. However, only a few samples revealed relevant relative abundances, while the majority of samples did not contain *Phaeomoniella* or *Eutypa*. Considering the number of *Pa. chlamydospora* spores on arthropod exoskeletons and the relative abundance of GTD pathogens detected during metabarcoding, we hypothesize that arthropod-mediated transmission is possible, but its relevance seems negligible and has to be considered in light of multiple influencing factors and other ways of spore transmission. Yet, our results highlight another mechanism of spore transmission emphasizing the importance of minimizing and protecting grapevine pruning wounds in order to prevent pathogen invasion.

## Figures and Tables

**Figure 1 jof-10-00237-f001:**
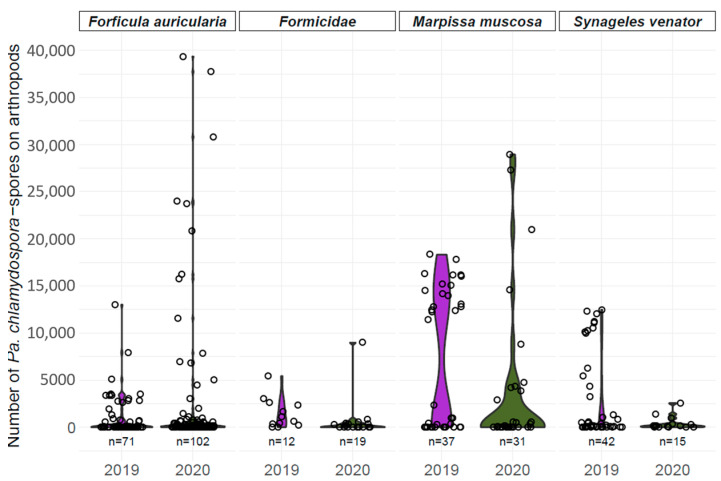
Spore loads of *Pa. chlamydospora* determined in 2019 (purple) and 2020 (green) on earwigs (*F. auricularia*), ants (Formicidae), and two jumping spiders (*M. muscosa* and *S. venator*) using qPCR. Arthropod sampling size (n) for each year and species is given below the individual violin plot.

**Figure 2 jof-10-00237-f002:**
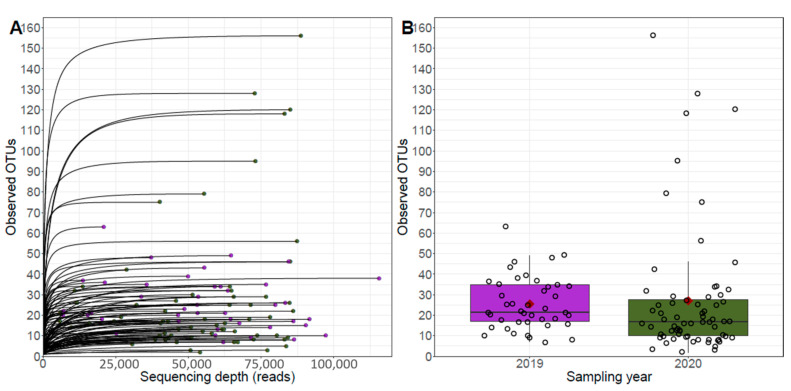
Alpha diversity metrics presenting the number of OTUs determined on earwig exoskeletons in the respective sampling year 2019 (purple) and 2020 (green). Rarefaction curves present the level of saturation of OTU detection (**A**), and boxplots show OTU richness in each year (**B**). Sampling sizes (n) were 42 in 2019 and 68 in 2020.

**Figure 3 jof-10-00237-f003:**
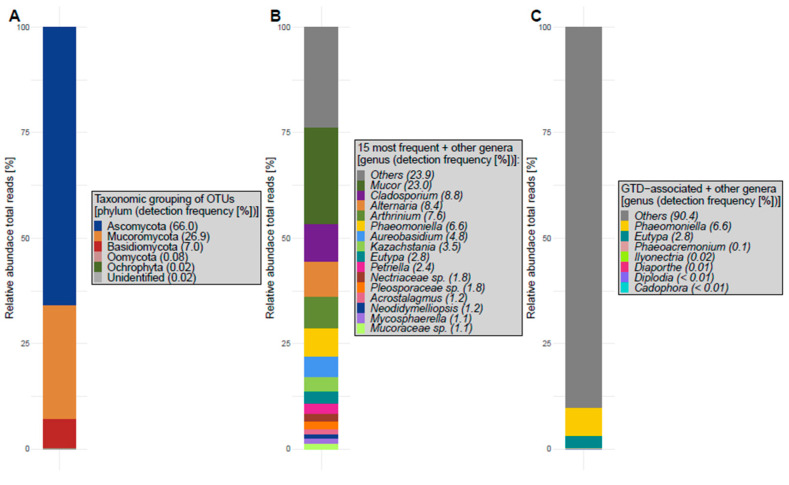
Mycobiome on earwig exoskeleton collected in the years 2019 and 2020 in two German vineyards. Bar plots of the relative abundance total reads (%) of phyla belonging to the kingdoms Fungi and Stramenopila identified (**A**), 15 most frequent genera (if genus-level assignment was possible) (**B**) and genera associated with GTDs (**C**). ‘Others’ are OTUs not included due to the respective filtering.

**Figure 4 jof-10-00237-f004:**
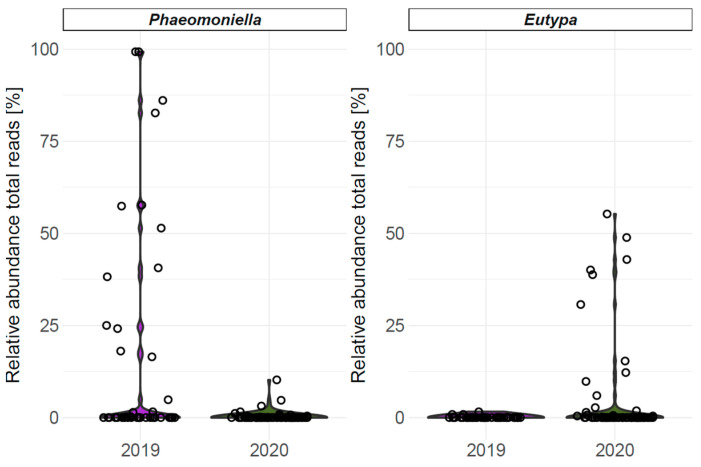
Relative abundance (total reads) [%] of OTUs assigned to the GTD-related genera *Phaeomoniella* and *Eutypa* found on earwig exoskeletons after mycobiome analysis, depending on the sampling year. Sampling sizes (n) were 42 in 2019 (purple) and 68 in 2020 (green).

## Data Availability

Raw sequence data have been submitted to the European Nucleotide Archive (https://www.ebi.ac.uk/ena/browser/home) with accession number PRJEB68302.

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
