# Peer review of "Arthropods as Vectors of Grapevine Trunk Disease Pathogens: Quantification of Phaeomoniella chlamydospora on Arthropods and Mycobiome Analysis of Earwig Exoskeletons"

_jof, 2024, doi:10.3390/jof10040237_

Round 1

Reviewer 1 Report

Grapevine trunk diseases is a very important, complex grapevine disease that is still poorly recognized. This manuscript presents very interesting research results carried out at two experimental vineyards in Germany. The authors determined the involvement of arthropods in the dissemination process of pathogens causing Grapevine Trunk Diseases. The authors quantify for the first time considerable spore loads via qPCR of the important pathogen Phaeomoniella chlamydospora on arthropods exoskeletons. New aspects include also the identification of mycobiome on earwig exoskeletons along with the analysis of their involvement in the dispersal of other pathogens of the grapevine diseases. All sections of the manuscript have been prepared very carefully. In a very interesting Discussion, the authors refer in detail to the results obtained, pointing to the possible role of arthropods in the epidemiology of a complex grapevine disease known as Grapevine Trunk Diseases. Knowledge about the mechanisms of infection and spread of many pathogens in the vineyard is so far very scarce. This manuscript should be published in JoF. The comments presented in Remarks should be taken into account.

Remarks

Title and Line 14 (and other places in text) - 'Arthropods as vectors of Grapevine Trunk Diseases' - this is not a precise term. Various organisms are vectors of disease causal agents (pathogens) and not of the disease itself (GTD pathogens would be more precise). – for consideration by the authors.

Line 72 ‘pathogenic material’ it is not precise, better use ‘pathogen inoculum’

Line 90 Pa. chlamydospora –the abbreviation Pa. should not be used, so maybe Pm. would be better: Phaeomoniella – m is the first letter differentiating this genus from Phaeoacremonium; Pa. should be used for Phaeoacremonium. I know that in other publications it is as the authors currently write. Probably should be left that way to avoid confusion

Line 103 'fungus-resistant' - would it be possible to clarify what specific fungus it is ?

Line 134 ‘A representative isolate’ – please indicate where this isolate from Culture Collection came from. In this study, no fungi were isolated on agar media.

Line 261 Formicidae – it needs correction

Line 281 it should be Mucoromycota instead of Mucormycota

Line 286 ‘of fungal phyla identified’ – it should be noted that Oomycota and Ochrophyta are not Fungi (it needs correction)

Line 293 Fig.3C. add a space

Line 374 it should be taken into account that Alternaria belongs to the Pleosporaceae family

Line 393 should be taken into account that Phomopsis is asexual stage of Diaporthe (Phomopsis should be remowed or write - Diaporthe/Phomopsis)

 Line 429 'arthropods cause pathogenic inoculum' ... this text requires correction 

Line 516 No. 14 Eur J Plant Pathol, Line 553 European Journal of Plant Pathology, Line 518 No. 15 Plant Dis. /with dot/ – the literature should be unified throughout whole  References

Line 523 ‘Phaeomoniella chlamydosporag’ – it needs correction

Line 636 it should be chlamydospora instead of chlamidospora

Author Response

Thank you very much for your remarks!

Reviewer 2 Report

The article explores further into the role of Forficula auricularia and other arthropods as vectors in the transmission of spores of Phaeomoniella chlamydospora and other pathogenic fungi that are causal agents of the grapevine trunk disease, and concludes that the contribution of arthropods to the spread of the disease is almost negligible, at least in experimental vineyards in Germany. The minimal role of these insects in spore transmission had already been demonstrated in previous studies. In this regard, the most significant finding seems to be that although previous studies have demonstrated the presence of fungi spores causing GTD and experimental transmission of the disease through arthropods in artificial transmission experiments, the spore load on arthropods collected from the field is much lower (several magnitude orders) than that used in artificial transmission experiments. Therefore, the involvement of these arthropods in disease transmission in the field is minimal, and this should be emphasized in the text and in the conclusions.

It is important to highlight the authors' effort in exploring the microbiome associated with the cuticle of insects. Although they found other pathogenic fungi related to the onset of esca and GTD besides Pa. chlamydospora and Eutypia, the relative levels of these are very low to be significant. Nevertheless, this is one of the first works that explore the involvement of Forficula auricularia in the dissemination of GTD using NGS (Next-Generation Sequencing).

It is also noteworthy the low detection of Pa. chlamydospora on insects using qPCR, and that these results are consistent with those obtained by NGS, especially considering that the samples used are positive for Pa. chlamydospora using nested-PCR. It would be worthwhile to discuss further regarding the rate of false positives obtained using nested-PCR as a routine detection method. Also, spores adhere to the insect cuticle through hydrophobic or electrostatic interactions, so using a solution with low concentrations of detergent or a saline solution could be more efficient for collecting spores than the water used in their extraction method. Perhaps the high variation observed in their spore determination is due to an inefficient and highly variable harvest of the spores present on the insect cuticle

Finally, the authors should review the last lines (455-457) of their conclusion; I do not see how the results of this study directly emphasize the importance of minimizing grapevine pruning wounds to minimize the risk of esca.

• Figure 3C: The "others" section (which corresponds to fungi not associated with GTDs) occupies the majority of the graph and does not allow for the appreciation of the proportion of fungi associated with GTDs. I suggest removing the "others" from this figure.

• Line 252: The fact that a Ct value is not obtained above the threshold does not mean that they do not contain spores; it means they are below the detection limit.

• Lines 392-397: The authors claim that the relative abundance of fungi associated with GTDs in Phaeomoniella is higher than the relative abundance of these same fungi found in grapevine leaves. They should specify the values of relative abundance against which they are comparing.

• Line 406: According to the authors, their NGS results agree with the qPCR results regarding only a few arthropod individuals being in contact with Pa. chlamydospora or Eutypa spores. Why don't they elaborate on the rate of false positives obtained through nested PCR?

• Line 425: According to their data, the contribution of arthropods to the dissemination of fungi causing GTDs is not "comparatively low"; it is negligible.

• The lower detection of Pa. chlamydospora using qPCR compared to detection using nested-PCR cannot solely be explained by the lower sensitivity of the former; nested-PCR can also yield a higher number of false positives because any contamination in the sample is magnified, particularly when it is carried out in two steps, as was done here.

• Annex A: It is very rare to see a correlation coefficient of 1.0 in a calibration curve of this type. Is that value correct?

Author Response

Thank your very much for your remarks!
